

# Robustness analysis in an inter-cities mobility network: modeling municipal, state and federal initiatives as failures and attacks toward SARS-CoV-2 containment

Vander L.S. Freitas[1], Gladston J.P. Moreira[1] and Leonardo B.L. Santos[2,3]

[1] Department of Computing, Universidade Federal de Ouro Preto, Ouro Preto, Minas Gerais, Brazil
[2] National Center for Monitoring and Early Warning of Natural Disasters, Sao Jose dos Campos, Sao Paulo, Brazil
[3] Department of Physics, Humboldt Universität Berlin, Berlin, Germany

Corresponding author
Vander L.S. Freitas,
vander.freitas@ufop.edu.br

## ABSTRACT

We present a robustness analysis of an inter-cities mobility complex network, motivated by the challenge of the COVID-19 pandemic and the seek for proper containment strategies. Brazilian data from 2016 are used to build a network with more than five thousand cities (nodes) and twenty-seven states with the edges representing the weekly flow of people between cities via terrestrial transports. Nodes are systematically isolated (removed from the network) either at random (failures) or guided by specific strategies (targeted attacks), and the impacts are assessed with three metrics: the number of components, the size of the giant component, and the total remaining flow of people. We propose strategies to identify which regions should be isolated first and their impact on people mobility. The results are compared with the so-called reactive strategy, which consists of isolating regions ordered by the date the first case of COVID-19 appeared. We assume that the nodes' failures abstract individual municipal and state initiatives that are independent and possess a certain level of unpredictability. Differently, the targeted attacks are related to centralized strategies led by the federal government in agreement with municipalities and states. Removing a node means completely restricting the mobility of people between the referred city/state and the rest of the network. Results reveal that random failures do not cause a high impact on mobility restraint, but the coordinated isolation of specific cities with targeted attacks is crucial to detach entire network areas and thus prevent spreading. Moreover, the targeted attacks perform better than the reactive strategy for the three analyzed robustness metrics.

## INTRODUCTION

Since early 2020, the SARS-CoV-2 quickly spread to the entire world and became a pandemic in a short time. As of 2 September 2020, the virus has reached more than 180 countries, with more than 26,065,382 confirmed cases of COVID-19, the disease caused by

the virus, and about 863,826 deaths, globally (*JHU Database, 2020*). In Brazil, there are more than 4,003,441 confirmed cases and nearly 123,926 deaths, with the first documented case located in the city of São Paulo on 25 February 2020 (*Cota, 2020*).

The design of containment strategies promoted in federal, state and municipal actions became an enormous challenge to prevent community transmission. In this context, the analysis of the inter-cities terrestrial mobility network is useful for decision making since the coordinated isolation of specific cities and states is crucial to spreading prevention.

The complex networks (*Estrada, 2012*) emerge as a natural mechanism to treat mobility data, taking areas as nodes and movements between origins and destinations as edges (*Barbosa et al., 2018*). A complex network is a graph (set of nodes and relations between them) that represents a complex system. A mobility network is a set of areas connected by the flow of people and, unlike physical networks (such as transportation infrastructures), they are social networks (*Santos et al., 2019a*).

The structure of the underlying network of a system reveals its ability to survive to random failures and coordinated attacks. Knowing which and how many nodes can be removed until the network completely fragments into small pieces is of great importance (*Barabási, 2016*). In this paper, we present a robustness analysis (*Barabási, 2016*; *Callaway et al., 2000*) on Brazilian mobility networks, motivated by the challenge of the COVID-19 pandemic and the seek for proper containment strategies. We propose strategies to identify which regions should be isolated first, their impact on people mobility, and how they compare to the so-called reactive strategy, which consists of isolating regions ordered by the date the first case of COVID-19 appeared.

We effectively damage the network structure through different strategies by systematically removing the cities (or states) that have more impact on mobility. Within the context of robustness analysis, a failure is the random removal of a node, and a targeted attack is the removal of a node based on a specific strategy. The local initiatives are here modeled as random failures because there is no central/global orchestration. It is possible that some cities (states) start to care about an epidemic before the others and/or before the country itself, either because their mayors (governors) have more political influence than the others, or due to local popular pressure. In both cases, the outcome for the city (state) is likely to diverge from the announced measures for the country at the federal level. Contrarily, the cooperation between cities, states, and the federal government characterize the targeted attacks, so that a federal level scheme guides the isolation process.

The present study employs the IBGE data from 2016 (*Instituto Brasileiro de Geografia e Estatística (IBGE), 2017*), which contains the flow of people between cities, considering only terrestrial vehicles from companies that sell tickets to passengers. Another data source, commonly used, is the pendular travels (*Brasil, 2020*) of people moving from home to work/study. Yet, the former is more recent and captures the flows of people between all pairs of Brazilian cities in a more general scenario. The data we use concerns the flow of people and does not cover the transport of supplies. The isolation of a region

consists of closing the borders to the flow of people to/from other regions, as performed in Wuhan, China. (*Li et al., 2020a*).

Our contributions are the robustness analysis of the Brazilian inter-cities mobility network, where random failures abstract local actions from cities or states, and the targeted attacks are the federal's. We assess the impacts of nodes' removal with three metrics: the size of the giant component, the number of components, and the total remaining flow within the network. Strategies based on centrality measures such as degree, betweenness, and topological vulnerability guide the targeted attacks. Lastly, we compare both the random failures and the targeted attacks with the reactive strategy. While the nodes' removals through targeted attacks follow the sorted values of the centrality measures, from the higher value to the lower, in the reactive strategy, the removal starts from the first node that notified COVID-19 in its territory, followed by the next, until the last node in a temporal order.

## MATERIALS AND METHODS

The complex network approach is often applied to treat mobility data, taking areas as nodes and movements between origins and destinations as edges. Formally, a network is defined as an undirected graph $G(V,E)$, consisting of the set $V$ of vertices (or nodes) and set $E$ of edges, with the total number of nodes $N = |V|$ and the total number of edges $|E|$. The edges' weights are represented as the matrix $W = \{w_{ij}\}$, for $i, j = 1, \ldots, N$, so that $w_{ij}$ is the weight between edges $i$ and $j$. The mean value and standard deviation of this matrix are $\overline{w}$ and $\sigma$, respectively.

The network flows (weights) (*Instituto Brasileiro de Geografia e Estatística (IBGE), 2017*) are here aggregated within the round trip, which means that the number of travels from city A to city B is the same as from B to A. We produce three types of undirected networks with a different number $N$ of nodes to capture actions in distinct scales (country and state):

1. $N = 5,420$—Brazil (BR): nodes are cities and edges are the flow of direct travels between them. The dataset encompasses almost all Brazilian cities.
2. $N = 620$—São Paulo state (SP): a subset of the previous network, containing only cities within the São Paulo state, the first Brazilian state with a confirmed case.
3. $N = 27$—Brazilian states (BS): in contrast with the others, in this network, each state is a node, and the edges are the accumulated flows between them.

Several networks are analyzed from the three models (BR, SP and BS), with flow thresholds employed in three levels: (i) original data with all recorded flow, (ii) only edges of at least an average flow, and (iii) a more restricted topology with the higher flows. The chosen thresholds are $\eta_0 = 0$, $\eta_1 = \overline{w}$ and $\eta_2 = \overline{w} + \sigma$. Edges with flows below these values are discarded. We thus end up with nine networks in total, as described in Table 1, where $N$ is the size of the network, and $|E|$ is the number of edges/links. The motivation behind the threshold levels is the fact that most centrality measures we investigated do not account for the flows and thus consider all edges with the same importance. Besides,
**Table 1 Networks' statistics.**

|  | Network | | |
|---|---|---|---|
|  | **BR** | **SP** | **BS** |
| $N = |V|$ | 5,420 | 620 | 27 |
| $\overline{w}$ | 48.04 | 73.20 | 2,032.29 |
| $\sigma$ | 100.21 | 122.79 | 4,397.86 |
| $|E|$ for $\eta_0$ | 65,264 | 9,592 | 474 |
| $|E|$ for $\eta_1$ | 15,505 | 2,610 | 108 |
| $|E|$ for $\eta_2$ | 4,217 | 758 | 44 |

Note:
The Brazilian (BR), Sao Paulo state (SP) and Brazilian states (BS) networks, with three flow thresholds: $\eta_0 = 0$, $\eta_1 = \overline{w}$ and $\eta_2 = \overline{w}+s$, where $\overline{w}$ is the average flow and s is the standard deviation.

neglecting some small flow connections may help to approximate the network measures to the real spreading dynamics of SARS-CoV-2 (*Freitas et al., 2020*).

## Measures of complex networks

The degree $k$ is the number of cities (or sates) that a city (state) is connected to, showing the number of possible destinations for the SARS-CoV-2. The betweenness centrality $b$ considers the entire network to depict the topological importance of a city in the routes more likely to be used. The vulnerability $\mathscr{V}$ accounts for the impact in the network efficiency when a particular city (state) is isolated. Lastly, the strength $s$ captures the total number of people that travel to (or come from) such places in a week. From a probability perspective, the cities that receive more flow of people are more vulnerable to SARS-CoV-2.

The topological degree $k$ of a node presents its connectivity: it is the number of edges it has to other nodes. The networks are undirected with no distinction between incoming and outgoing edges. On the other hand, the betweenness centrality captures the importance of a node. Between any pairs of nodes $l$ and $m$ of a connected network, there is at least one shortest path, and the betweenness $b_i$ is the rate of such paths that pass through $i$ (*Barthélemy, 2004*):

$$b_i = \sum_{l \neq m \neq i} \frac{g_{lm}(i)}{g_{lm}}, \tag{1}$$

in which $l, m, i \in V$, $g_{lm}$ is the total number of shortest paths (or geodesic paths) between $l$ and $m$, and $g_{lm}(i)$ are those that pass through $i$.

The efficiency $e_{ij}$ in the communication between a pair of nodes $i$ and $j$ can be defined as the inverse of the shortest path length between them, and the network efficiency $\mathscr{E}$ (*Goldshtein, Koganov & Surdutovich, 2004*; *Wang, Du & Deng, 2017*) is

$$\mathscr{E} = \frac{\sum_{i \neq j} e_{ij}}{N(N-1)}, \tag{2}$$

the average of all efficiencies, with $i, j \in V$. The vulnerability index $\mathscr{V}_i$ (*Santos et al., 2019b*), quantifies how vulnerable to the removal of node $i$ a network is:

$$\mathscr{V}_i = \frac{\mathscr{E} - \mathscr{E}_i^*}{\mathscr{E}}, \tag{3}$$

in which $\mathscr{E}_i^*$ is the average network efficiency after the removal of node $i$. In brief, the flow of information is considered more efficient in networks with small shortest path lengths.

The strength $s_i$ of a node is the accumulated flow from incident edges:

$$s_i = \sum_{j=1}^{N} w_{ij}. \tag{4}$$

## Robustness

The robustness of a network is its capacity to keep connected even after the removal of nodes and/or edges (*Barabási, 2016*). A breakdown (for example, an energy drop) of some computers in computer networks, or a car accident on an important road, are usually unpredictable events that depend on several internal and/or external causes, thus characterizing a system failure. Conversely, an intentionally removed node to disrupt the network structure typifies an attack (*Schneider et al., 2011*). We propose strategies to identify the municipalities (states) that play a key role in mobility. Our motivation is the fact that real networks are robust to random failures but are fragile to attacks (*Barabási, 2016*; *Callaway et al., 2000*; *Cohen et al., 2000*; *Iyer et al., 2013*). The main question is to figure out how many and which nodes must be removed until the network collapses. Understanding which cities are important for mobility to know exactly which node to isolate in a disease outbreak is of major interest.

We keep track of three measures to quantify the network response to both random failures and targeted attacks when a rate $f$ of nodes are removed: the number of nodes in the giant component $P_\infty(f)$, the total number of components $C(f)$, and the total remaining flow $\|W\|(f) = \sum_{ij} w_{ij}$. Within this framework, whether a single node or a small group is isolated from the rest, it is considered a component itself.

There are different ways to choose which node to remove. Random failures are the trivial case for which nodes are randomly selected. However, targeted attacks demand some strategy like always removing the nodes with higher degrees. We propose four strategies: deleting nodes with a higher degree (max $k$), betweenness (max $b$), vulnerability (max $\mathscr{V}$), and strength (max $s$). Attacks oriented by higher degrees are effective to reduce the size of the giant component and produce better results than non-local measures in most cases (*Iyer et al., 2013*).

The BR network ($N = 5{,}420$) has a degree distribution that follows a power-law with a coefficient $\gamma = 2.57$, which characterizes a scale-free topology. This means that, under random failures, the critical threshold $f_c = 0.9911$, for $f_c = 1 - (1/(\kappa - 1))$ with $\kappa = \langle k^2 \rangle / \langle k \rangle$, gives the exact fraction of random 155 node removals that break the network. This structure is strongly robust to failures, that is, almost all nodes must be removed before the giant component takes apart (*Barabási, 2016*). On the other hand, such networks are vulnerable to attacks, especially when they target higher degree nodes (hubs).
**Table 2 Networks' measures.**

| | Network | | | | | | | | |
|---|---|---|---|---|---|---|---|---|---|
| | $\eta_0$ | | | $\eta_1$ | | | $\eta_2$ | | |
| | BR | SP | BS | BR | SP | BS | BR | SP | BS |
| $\langle k \rangle$ | 24.08 | 15.47 | 17.56 | 5.72 | 4.21 | 4.0 | 1.56 | 1.22 | 1.63 |
| $\langle b \rangle$ | 5,574.09 | 504.24 | 4.56 | 4,828.08 | 397.67 | 13.04 | 2,177.91 | 125.7 | 6.41 |
| $\langle s \rangle$ | 1,156.86 | 1,132.35 | 35,677.91 | 845.23 | 813.66 | 30,162.76 | 504.93 | 473.9 | 21,043.46 |
| $\langle \mathcal{V} \rangle$ | 4.18E−4 | 3.62E−3 | 7.57E−2 | 4.97E−4 | 4.6E−3 | 8.11E−2 | 6.95E−4 | 6.53E−3 | 0.1 |

Note:
Average degree $\langle k \rangle$, average betweenness $\langle b \rangle$, average strength $\langle s \rangle$ and average vulnerability $\langle \mathcal{V} \rangle$ for the Brazilian (BR), Sao Paulo state (SP) and Brazilian states (BS) networks under flow thresholds $\eta_0$, $\eta_1$ and $\eta_2$.

Robustness is measured by *Iyer et al. (2013)*

$$R = \frac{1}{N}\sum_{i=1}^{N}\frac{\Gamma(i/N)}{\Gamma(0)}, \tag{5}$$

for $R \in (0, 1/2)$ and $\Gamma(f)$ is the network response function after removing a fraction $f$ of its nodes. The higher the $R$, the more robust the network is according to the function $\Gamma$, which could be either $P_\infty$ or $\|W\|$. Note that the normalization factor $1/N$ allows the comparison of networks of different sizes. For $P_\infty$, the star-like topology reaches the minimum value $R = 1/N$, and the complete graph achieves the maximum

$$R = \frac{1}{2}(1 - 1/N).$$

The $R$ measure cannot be computed from $C(f)$, since this function does not always decrease like in $P_\infty$ and $\|W\|$. The number of components and their number of participants may oscillate instead. Two components with dozens of nodes each or two components with a single node are evaluated alike with $C(f)$, thus not giving a direct notion of connectivity or flow.

The simulations of the next section were carried on an Intel(R) Core(TM) i5-4210U CPU 1.70 GHz × 4, with 8 GB Ram, using Python programming language. The respective data and source code are available at https://github.com/vanderfreitas/network_ robustness.

## RESULTS

The measures related to the BR, BS, and SP networks for each flow threshold are summarized in Table 2 and Fig. 1 presents a sketch of the national network with two different flow thresholds.

All measures (degree, betweenness, vulnerability and strength) for the BR network under $\eta_0$ exhibit the cities of São Paulo and Belo Horizonte within the top-five higher values and most present Campinas and Brasília. Concerning the SP network, the measures rank the cities of São Paulo, Campinas, São José do Rio Preto, and Ribeirão Preto within the top-five values as well. Differently, the BS network does not display a clear pattern for the degrees, but the states of São Paulo and Minas Gerais come out in the first positions
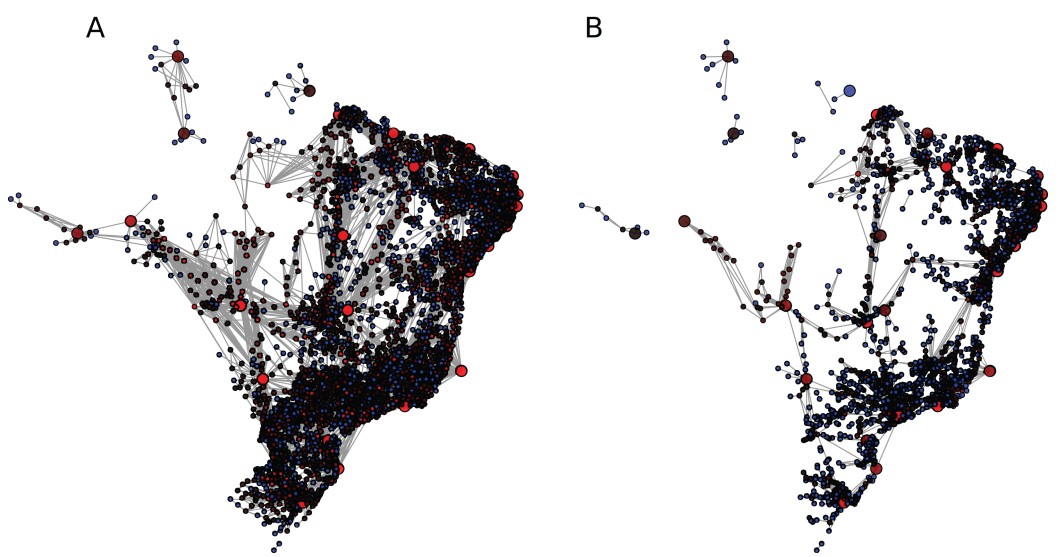

**Figure 1** **Brazilian mobility network (BR) under (A) $\eta_1$ and (B) $\eta_2$.** The larger nodes are state capitals. Nodes with smaller degrees are blue, with higher degrees are red and intermediate values are dark red. The figure for $\eta_0$ is not properly visible due to its 65,254 edges.

for betweenness, vulnerability and strength (see the corresponding Tables in the Supplemental Material).

The reactive strategy (COVID-19 curve of Fig. 2) reaches an intermediate performance, with both $R$ values and $C$ curves amid random failures and targeted attacks. Despite not better, the results for the reactive strategy are comparable to the targeted attacks when the remaining flow $\|W\|(f)$ is at stake (bottom of Fig. 2). However, the targeted attacks are more effective when it comes to $P_\infty(f)$.

We normalized $C(f)$ according to the initial number of components (before removing nodes). There is about 25 times the number of components observed in $C(0)$, when half of the nodes ($f = 0.5$) are removed under the guidance of the betweenness centrality in the BR network with $\eta_0$ (blue curve of Fig. 2A). The $C(0)$ does not equal 1 (one) necessarily since some cities are either isolated or compose small components that do not have terrestrial flows of people to the rest.

The number of components $C(f)$ increases almost linearly under random failures for BR with $\eta_0$ (black curve of Fig. 2A) and only decreases in the end, with $f \approx 0.9$. The giant component for the same network is initially well connected and does not break easily, then the number of components remains closely the same. On the other hand, $C(f)$ only decreases for $\eta_1$ and $\eta_2$, due to the lower number of links. This results in a maximum number of components that is smaller than in $\eta_0$, since the initial number of clusters is higher in the former cases. The same is observed in Figs. 3 and 4 for SP and BS network, respectively.

Attack-wise, the degree is more well-succeeded in decreasing the size of the giant component, and strength performs better regarding the total remaining flow in both BR and SP networks. The degree (yellow curves) indeed decreases the size of the components,

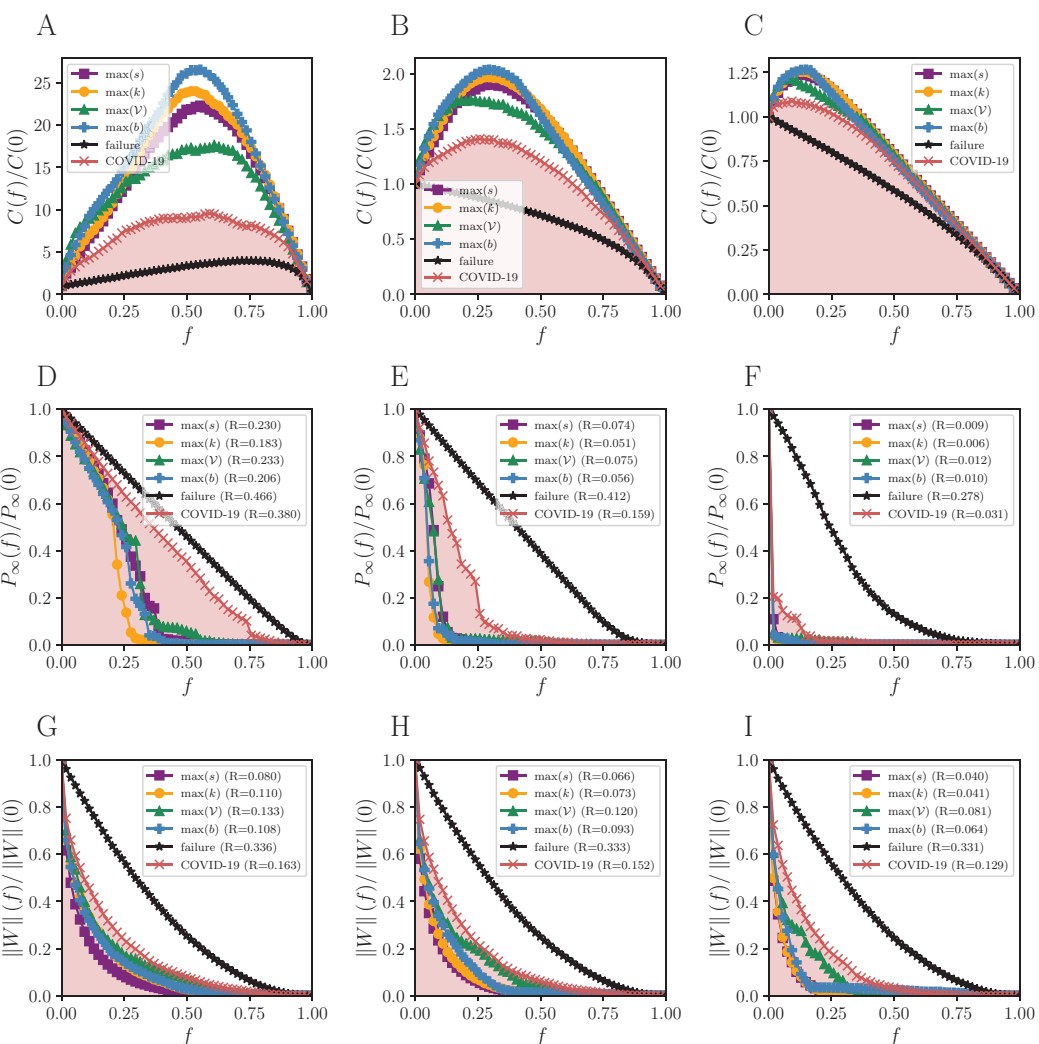

**Figure 2 Robustness analysis for the Brazilian mobility network (BR).** Attack strategies: max *s* (strength), max *k* (degree), max $\mathcal{V}$ (vulnerability), max *b* (betweenness), and the reactive (COVID-19 curve). The failure curve is the average behavior for 50 random simulations. Three connection thresholds are considered: (A, D and G) $\eta_0$; (B, E and H) $\eta_1$ and (C, F and I) $\eta_2$, as in Table 1. Functions to evaluate the impact of removing a fraction *f* of nodes: the normalized number of connected components $C(f)/C(0)$, the normalized size of the giant component $P_\infty(f)/P_\infty(0)$, and the normalized remaining flow in the system $||W||(f)/||W||(0)$.          

because it targets the most connected nodes. The betweenness, on the other hand, generates a larger number of components (blue curves), since it detects the shortest paths between groups of well-connected nodes, which coincide with their bridges. Some methods for community detection - like the Girvan-Newman—systematically remove the edges with higher betweenness (*Easley & Kleinberg, 2010*).

Although some cities have not reported COVID-19 cases until 2 September 2020, we computed the *R* related to the reactive strategy, since the number of remaining cities is negligible: 61 for BR (1.1% of its nodes). Note that Eq. (5) takes into account the entire curve of $\Gamma(f)$ for $f \in [1/N, 1]$. We verified that the remaining nodes of the BR network would impact in fluctuations of a maximum of $10^{-2}$ in *R* for $P_\infty(f)$, and $10^{-3}$ for $||W||(f)$.

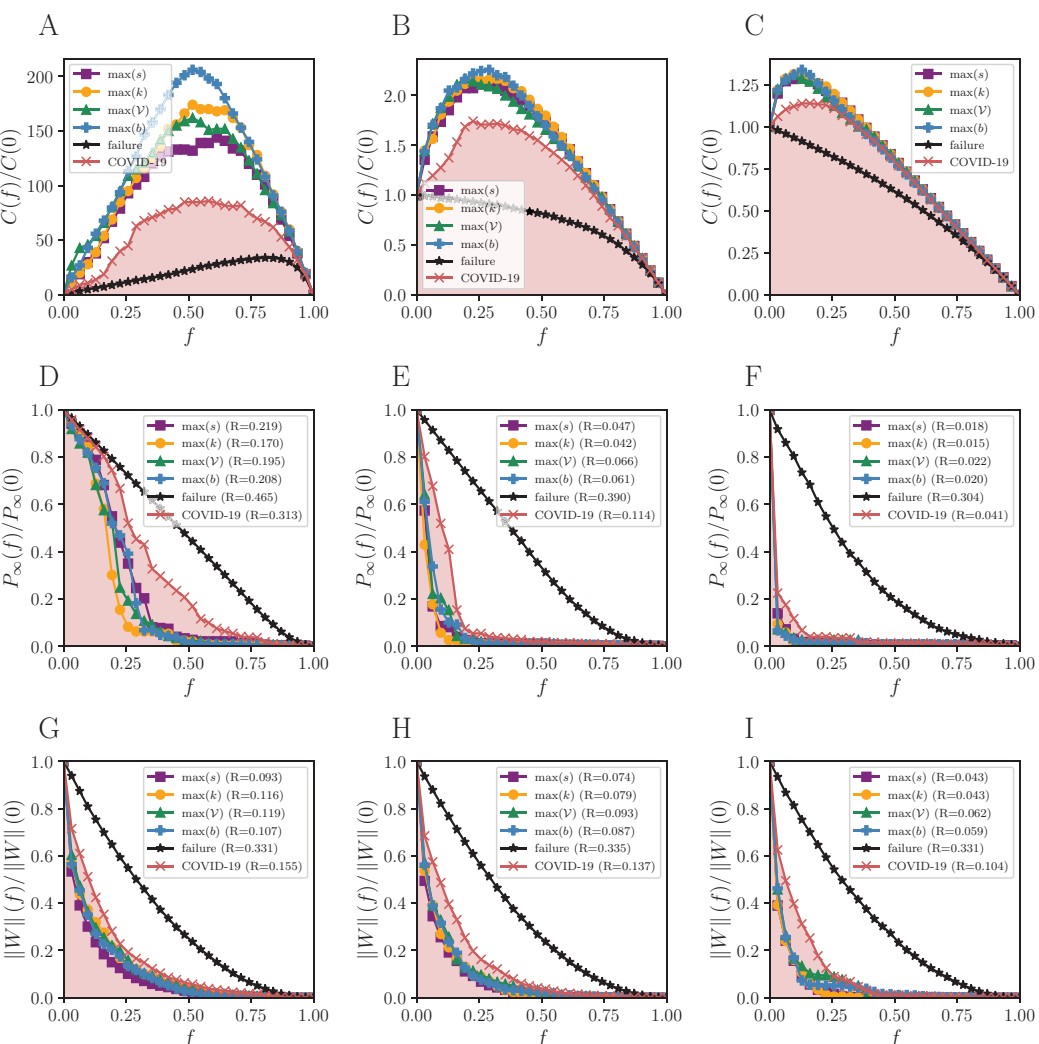

**Figure 3 Robustness analysis for the Sao Paulo mobility network (SP).** Attack strategies: max *s* (strength), max *k* (degree), max $\mathcal{V}$ (vulnerability), max *b* (betweenness), and the reactive (COVID-19 curve). The failure curve is the average behavior for 50 random simulations. Three connection thresholds are considered: (A, D and G) $\eta_0$; (B, E and H) $\eta_1$ and (C, F and I) $\eta_2$, as in Table 1. Functions to evaluate the impact of removing a fraction *f* of nodes: the normalized number of connected components $C(f)/C(0)$, the normalized size of the giant component $P_\infty(f)/P_\infty(0)$, and the normalized remaining flow in the system $||W||(f)/||W||(0)$.

The reactive strategy has a low impact on the number of connected cities in the giant component, but has a strong effect in the remaining flow in BR. There is an important feedback mechanism in this case: the emergence of COVID-19 cases is possibly associated with both imported cases and community transmission between cities in the country. Thus, the flow of people is on both sides of this relation.

The São Paulo mobility network (SP) produces similar results as the BR, but the topological vulnerability starts to play a more significant role than in BR, being the second-best under $\eta_0$ and $P_\infty$.

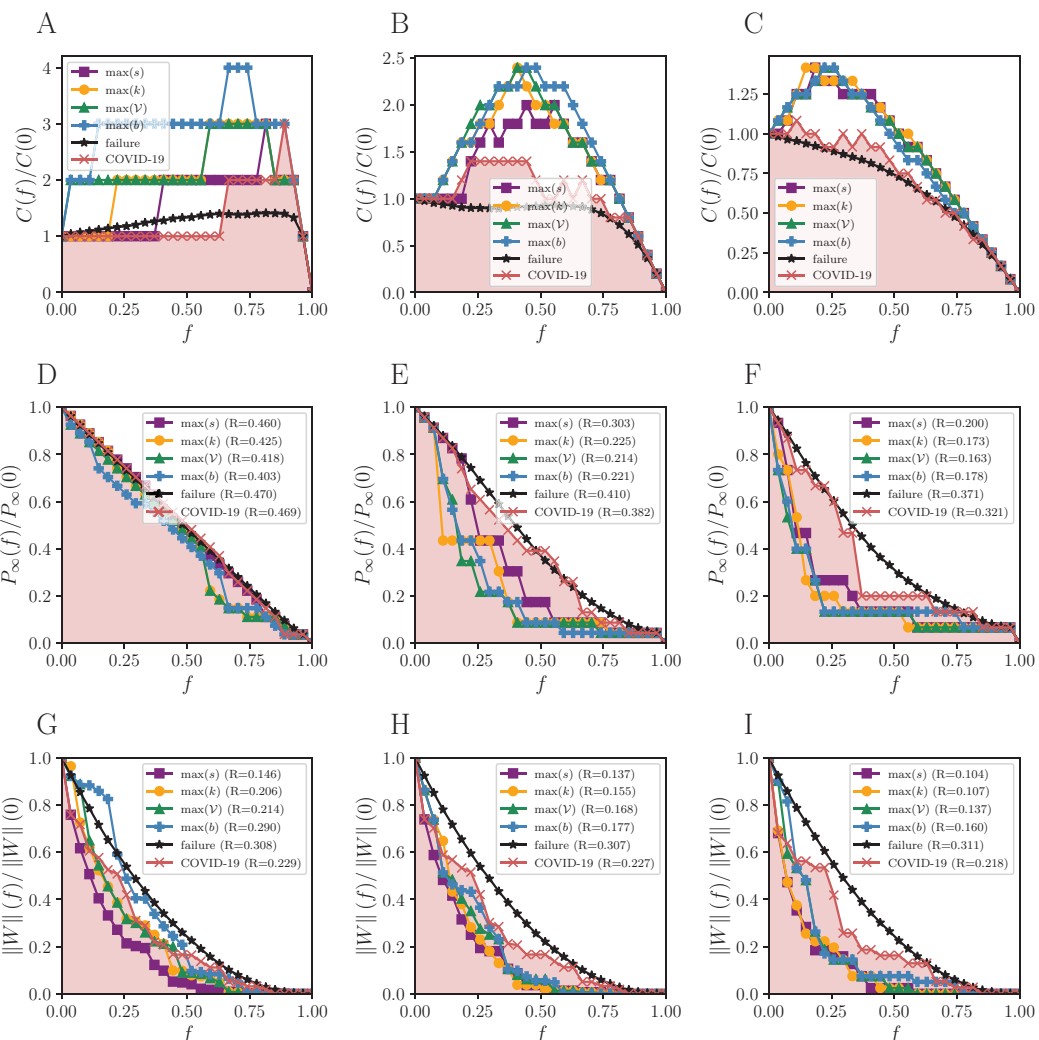

**Figure 4 Robustness analysis for the Brazilian states' mobility network (BS).** Attack strategies: max $s$ (strength), max $k$ (degree), max $\mathcal{V}$ (vulnerability), max $b$ (betweenness), and the reactive (COVID-19 curve). The failure curve is the average behavior for 50 random simulations. Three connection thresholds are considered: (A, D and G) $\eta_0$; (B, E and H) $\eta_1$; and (C, F and I) $\eta_2$, as in Table 1. Functions to evaluate the impact of removing a fraction $f$ of nodes: the normalized number of connected components $C(f)/C(0)$, the normalized size of the giant component $P_\infty(f)/P_\infty(0)$, and the normalized remaining flow in the system $||W||(f)/||W||(0)$.               

The differences between failures and attacks are only noticeable for higher thresholds in the network formed by the Brazilian states (BS)—see Fig. 4. Removing nodes with the attacking strategies does not cause much more impact than picking by chance under $\eta_0$ and $P_\infty$. The results differ for other thresholds when the shortest paths between nodes increase.

Notice that some plateaus represent regions where the removal of nodes does not impact on robustness. An example is the interval $f \in [0.2, 0.75]$ of Fig. 4F, where attacking nodes under the betweenness guidance does not cause any harm, because the referred nodes do not belong to the giant component. Interestingly, the attacks and failures perform

similarly, and sometimes the failures are even more effective (Figs. 4E and 4G). The strategies follow the same order of efficacy for $\|W\|$ under all thresholds: strength, degree, vulnerability, and betweenness, with strength being the best and betweenness the worst. The reactive strategy is even better than betweenness for $\eta_0$.

Regarding $P_\infty$, there is an increasing importance of the vulnerability measure from BR to BS. Besides, while the degree is the best measure to guide the attacks for the National and São Paulo networks, it is not for the BS, where vulnerability and betweenness have more importance. Similarly, in BR and SP, for $\|W\|$, the strength is the leading measure for attacks, and vulnerability is the worst. Conversely, although strength is also the best for BS, betweenness is the worst.

## DISCUSSION

As expected (*Barabási, 2016*), random failures do not break the network until almost all nodes are removed, due to its scale-free structure, and all targeted attacks dismantle the networks for small *f*, except for the reactive strategy. The higher the threshold, the fewer nodes must be removed to break the network structure since the giant component is initially smaller than the observed for $\eta_0$. The *R* measure shows that the more effective attack strategy for $P_\infty$ is guided by degree, and by strength for $\|W\|$ for all thresholds. The smaller the *R*, the more destructive the corresponding attack strategy is. The maximum number of components arises in targeted attacks guided by the betweenness centrality for BR and SP networks. When it comes to the BS network, the same happens for $\eta_0$, but other measures also hit the maximum for other thresholds.

The reactive strategy produces an impact similar to that of targeted attacks on decreasing the flow of people, although slightly worse. The number of remaining connected cities is always higher. Therefore, despite reacting to the disease spreading is a valid action, targeted attacks provide better results in terms of the size of the giant component and remaining flow in the system.

The cities from the state of São Paulo that have higher values are also cited in recent studies (*Freitas et al., 2020*; *Guimarães et al., 2020*) on the most vulnerable cities to COVID-19 due to the intensive traffic of people.

Quickly breaking the transmission network is vital to contain any highly contagious disease, which demands the rapid implementation of control measures such as travel restrictions. Cities that preemptively adhered to the measures reported fewer cases than the others, and the virus reached them later (*Tian et al., 2020*). The city of Wuhan was the main focus in China, and the complete isolation of the area was essential to mitigate the virus spreading (*Li et al., 2020a*). On the other hand, the rest of the world received the SARS-CoV-2 concurrently at different places and had to divide efforts to restrain it.

The targeted attacks are especially relevant in areas where people are not sufficiently tested for COVID-19 since the reactive strategy strongly depends on effective epidemic surveillance. *Li et al. (2020b)* estimate that 86% of infections were undocumented in China before the travel restrictions of 23 January 2020, and the undocumented infections were the source of 79% of the documented cases. Underreporting is also present in

Brazil, as *Hallal et al. (2020)* pointed out in a nationwide seroprevalence survey they conducted.

The US response to COVID-19 is mostly guided by Governors and Mayors primarily because of their political system. Korea and Taiwan implemented a centralized national strategy with the support of other government instances (*Haffajee & Mello, 2020*; *Kim, Oh & Wang, 2020*). Canada has the Health Portfolio Operations Centre (HPOC), which concentrates the operations at different levels of government. While in UK the response to the crisis was diverse in the different regions, in Switzerland, the communication and agreement between all levels of government was strong from the very beginning, based on mutual learning and integration (*Gaskell & Stoker, 2020*). The *Organisation for Economic Co-operation & Development OECD (2020)* argues that coordinated response across regions and states minimize coordination failures since they avoid the "pass the buck" behavior.

We assume that the targeted attacks represent the centralization of the efforts to isolate municipalities and states following specific and well-engineered strategies. Such coordination is only possible with a consensus at federal, state, and municipal levels since systematized isolation rely on the adherence of all involved parts. The random failures, on the other hand, abstract independent and decentralized actions. Within this framework, the federal initiatives towards SARS-CoV-2 containment are more effective in breaking the transmission network than leaving the cities (or states) on their own. We have shown that random failures usually take longer to dismantle the networks than choosing the nodes with some criteria.

However, both the targeted attacks and the reactive strategy are possibly not feasible in some regions due to the widely divergent kinds of issues they may face (*Gaskell & Stoker, 2020*), where the authorities must tailor specific strategies at the local level. The conducted robustness analysis points out the more central cities according to the network metrics and how their isolation impacts in connectivity and the flow of people. We thus present action plans that depend on cooperation and could conceivably rearrange in real-world scenarios.

## CONCLUSIONS

Based on the robustness analysis, the reactive strategy is not effective in reducing the size of the giant component nor breaking the mobility network into disconnected groups when compared to the targeted attacks. Moreover, the federal actions have a substantial impact on the network, while the local ones usually do not break it before almost all cities are isolated. Choosing the cities with higher degrees for the targeted attacks is the best option in most cases, considering the size of the largest component, especially for the two largest networks ($N = 5,420$ and $N = 620$ cities). However, there is a transition, showing that the vulnerability index performs nearly the same as the degree for the São Paulo State network, and it is the best choice for the network of the Brazilian states ($N = 27$ nodes) under most threshold levels. The total flow of the network is affected similarly by both the targeted attacks and the reactive strategy, but the former is more well succeeded when guided by the strength measure. Lastly, the removal of regions ordered by

their betweenness centrality generates a higher number of disconnected islands in the mobility network, which ensures the containment of the disease within small isolated groups.

## ACKNOWLEDGEMENTS

We thank Catia S.N. Sepetauskas, Jeferson F. Mendes, Jussara Angelo, and Thais C.R.O. Konstantyner for the valuable discussions. Besides, we appreciate the reviewers and the editor for their detailed examination of our manuscript, which helped us to improve important aspects of the presentation.

### Funding

This work was supported by the Sao Paulo Research Foundation (FAPESP), Grant Numbers 2015/50122-0 and 2018/06205-7; DFG-IRTG Grant Number 1740/2; CNPq Grant Number 420338/2018-7; CAPES Grant Number 23038.014333/2020-46. The funders had no role in study design, data collection and analysis, decision to publish, or preparation of the manuscript.

### Grant Disclosures

The following grant information was disclosed by the authors:
Sao Paulo Research Foundation (FAPESP): 2015/50122-0 and 2018/06205-7.
DFG-IRTG: 1740/2.
CNPq: 420338/2018-7.
CAPES: 23038.014333/2020-46.

### Competing Interests

The authors declare that they have no competing interests.

### Author Contributions

- Vander L.S. Freitas conceived and designed the experiments, performed the experiments, analyzed the data, prepared figures and/or tables, authored or reviewed drafts of the paper, and approved the final draft.
- Gladston J.P. Moreira analyzed the data, prepared figures and/or tables, authored or reviewed drafts of the paper, and approved the final draft.
- Leonardo B.L. Santos conceived and designed the experiments, performed the experiments, analyzed the data, prepared figures and/or tables, authored or reviewed drafts of the paper, and approved the final draft.

### Data Availability

Data and source code are available at https://github.com/vanderfreitas/network_robustness

## Supplemental Information

Supplemental information for this article can be found online at http://dx.doi.org/10.7717/peerj.10287#supplemental-information.

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
