# Peer review of "Robustness analysis in an inter-cities mobility network: modeling municipal, state and federal initiatives as failures and attacks toward SARS-CoV-2 containment"

_PeerJ, doi:10.7717/peerj.10287_

## Round 0.1 · original submission · Major Revisions

I apologize for the delay in sending this decision. I was able to secure two good reviews of your manuscript. It's my opinion that authors must do a much better job in improving the flow of ideas throughout the whole paper. The way to structure paragraphs is not adequate, you very often split sentences about the same subject into two or more paragraphs. You also keep restating the motivation of the study over and over again in several parts of the Methods, this makes the reading a little bit chunky. Try to improve the flow of ideas. Almost all Results section need to be reformulated for clarity. Avoid repeating what is already written in the figure and table legends. Instead, call attention to any specific detail or summarize the pattern shown in them, and cite them at the end of the sentence. The conclusions section actually doesn't provide conclusions and need to be reformulated entirely. See my comments on that in the PDF attached. English language needs to be carefully revised. Also, pay special attention to R1 suggestion on how to improve figures.

Also, you definitely need to clarify the assumptions for the working hypothesis and make them understandable to a broader audience. What do you mean by attack and failure in this context?

I second R2 about the necessity to provide raw data used for analysis. Provide more details in the M&M section that allow people to reproduce your analysis. For example, which software was used for analysing the data?

Overall, the manuscript lacks citations, for the metrics (and formulae) you used, some concepts, as a support for assumptions,
I have made several comments and corrections directly in the attached PDF. Please, remember to refer to it when revising your manuscript.

Reviewer 1 ·

Basic reporting

The authors present a network analysis of an inter-cities mobility network based on Brazilian data from 2016 in order to describe the government initiatives concerning SARS-CoV-2 strategy measures of containment. The manuscript is clear, well-structured, with relevant results illustrated by Figures. Data comes form IBGE – Instituto Brasileiro de Geografia e Estatística.

In relation to the figures, I have two comments: the visualization of Figure 1 may be improved using blue color for the nodes with smaller degrees and light gray for the links; the captions of Figures 2, 3 and 4 should include that the procedure of removed nodes is based on different strategies as described on “Robustness” sub-section (deleting nodes with a higher degree, vulnerability, betweenness and strength). Moreover it is not clear in the caption how the COVID-19 cases (what means “temporal sequence”?) and “failure” curves are constructed; it is not clear the meaning of giant component and remaining flow for those curves. Finally I did not figure out the reason that there is a value of robustness measure for COVID-19 curve in Figure 4 but not in Figures 2 and 3.


I think that the authors should present in a supplementary material a more complete list of results obtained from their analysis. For instance, in discussion section, the authors refer the two cities (or states) with the higher values of network measures in relation to the 3 selected networks: Brazilian cities (BR), Brazilian states (BS) and São Paulo state (SP). I suggest the authors include a complete list (in supplementary material) that reveal the order of states (BS) in relation to the higher values of the considered network measures; for BR network, the authors may select the capitals of the states, and for SP network, they may select, for example, 30 cities with highest number of cases.

Experimental design

The authors set up a methodological procedure assuming the local initiatives as failures and the federal initiatives as attacks on the networks. I have some questions related to those assumptions as well as some conclusions associated with the obtained results.

First of all, it would be reasonable to provide an explanation to those assumptions; for instance, in conclusions section the authors affirm that the attacks are isolation measures determined by the federal government and applied at a municipality level. As far as I know, there is a legal debate in Brazil about the assignment in relation to isolation of the municipalities. Is there a final decision about it? In case of affirmative answer, I suggest to include a reference about it; in case of negative answer, it would be interesting to justify the assumptions.

In order to highlight the results, it is essential to clarify how the disease information concerning the temporal sequence of COVID-19 cases is related to the procedure of removed nodes that is fundamental for the methodology.

Validity of the findings

The authors perform a comparative analysis of the network measures, pointing out which one is better or worse for each network. In general they conclude that the strength is a more suitable measure to guide the attacks due to the smaller value of R. As far as I understood, the attack is more effective than the failure for BR and SP networks since the curves fit better for the flow than for the giant component; however, both of them fits well for BS network. Do the authors agree with that summarization? Does that result provide any suggestion about the relevance of the strategies of the state government strategies?


Undoubtedly the findings are relevant since it assigns a robustness analysis based on mobility network of a large country with a significative number of COVID-19 cases. The analysis may be useful for other transmitted diseases in the country. However, since it is a complex system, other factors get influence in the government decision of isolation; for instance it is very hard to isolate some large cities that are essential for the distribution of food and drugs. In this sense, it is not easy to determine which cities should be isolated. How do the authors make compatible their findings with that actual scenario?

Additional comments

I think that the manuscript may be published in PeerJ after clarifying the arisen points that I have classified as minor revision, since I suppose the authors had obtained the results suggested for supplementary material.

Reviewer 2 ·

Basic reporting

The authors report a timely study of a Brazilian transport network to identify potential strategies assisted by network analyses to contain the spread of Covid-19. The manuscript is clear, structured and easy to read. However, some notations could be more intuitively defined, for instance, why are the number of nodes in the giant component called P∞( f ), I don’t understand what infinite has to do with it? More importantly, regarding the reproducibility of the results and access to raw data, I did not find any Git repository or an open server where the data and associated scripts to reproduce the results are deposited. I strongly suggest the authors to make their code and data available in an open platform.

Experimental design

Overall, I am convinced with the technicality of the analyses design. However, a main concern I have is using the size of giant component to assess the performance of the targeted attack. May be a complimentary metric such as the actual number of connected components (or clusters) for each node removal might indicate how the network is effectively destabilized, even when the giant component is intact. This might mean certain transportation among major cities are very strongly connected, however, many rural cities could be disconnected. So, in addition to the 2 metrics the authors have used to assess the network performance after node perturbations, I would like to see a metric that captures how many components or clusters remain after each perturbation. This might give a picture of how the cities are actually disconnected after node removals. I can see that fraction of remaining flow within the network could be useful but, still I suspect a few large components in the network might dominate the flow.
Another strategy to choose node(s) for removal could be to employ minimum cut sets? i.e. the idea is to identify a minimal sets of nodes that disrupt the entire (or major) topology of the network? More details about this could be found here: Klamt and Gilles, 2004, Bioinformatics

Validity of the findings

Although I could partly agree with the author’s claim that global (or Federal) actions are more effective compared to local (or municipal) actions to disrupt the transportation network, I am not entirely sure if this can be concluded from the results presented as it is. As I mentioned in the earlier paragraph, I would like to see more complementary metrics to assess the network disruption both locally and globally. And may be an intermediate strategy where both local and global policies must be coordinated could also emerge if more metrics are included in the study.

---

## Round 0.2 · Minor Revisions

Please, pay attention to R1 comments on some specific aspects of the manuscript that still deserve the authors' attention. I second their claim about the attack/failure terms that still appear in the abstract and are extremely confusing.

Reviewer 1 ·

Basic reporting

The authors present a revised version of the manuscript with huge changes of the manuscript related to the text structure and, at least some added results since they extend the period of their analysis until September 2020. In some sense it is a “new” manuscript and due to that, in my point of view, it would be reasonable the authors point out, at least, the number of lines where they have introduced the changes that were required by us in the previous report. Following the tracked-changes version of the manuscript, I tried to catch the required changes. Another relevant point is to highlight in the cover letter the implications of extending the time interval of analysis for the discussion of the results. Summarizing the authors tried to answer the editor's criticisms about the text – that was relevant for improving the quality of the manuscript – but do not take care to make clear in the letter what was done point by point, at least when the changes are related to the content of the manuscript (for isntance, it was not necessary to do it for English revision).

The authors have improved the quality of the figures and its description in figure captions. Now, based on an extended data bank (until September), it is possible to a value of robustness measure for COVID-19 curve in Figures 2 and 3 as in Figure 4.

As I have suggested, the authors also present in a supplementary material a detailed analysis that is useful for reproduction as well as relevant for comparison with other scenarios.

Experimental design

Again, I am not quite sure whether, how and where (which lines???) the answer for that question (presented in the letter) was introduced in the manuscript.

Reviewer: In order to highlight the results, it is essential to clarify how the disease information concerning the temporal sequence of COVID-19 cases is related to the procedure of removed nodes that is fundamental for the methodology.

Authors: We improved the explanation within the manuscript. The reactive strategy consists of the removal of cities/nodes according to the date they documented the first case of COVID-19 (see Comment #1). This ordering may not impact the mobility like attacking the nodes according to their corresponding centrality measures (targeted attacks - starting with the node with higher centrality value towards the nodes with smaller). Three metrics evaluate the impact of the reactive strategy and the targeted attacks in mobility: the number of components, the size of the giant component, and the total remaining flow of people.

Validity of the findings

Concerning the summarization of the results, do the authors summarize it as below (which lines??)

Authors: "Attacks usually cause more impact on the networks than both the reactive strategy and random failures. The only exception is when one evaluates the size of the giant component in the BS network under η0, where all the strategies perform roughly the same. We observed that attacks guided by strength are more effective to diminish the total remaining flow of people; those guided by degree usually result in smaller giant components and; the ones guided by betweenness generate more isolated components."

In my point of view, it is reasonable to explain to the reviewers whether and which results were changed taking into account a longer time interval besides the value of R in Figure 2 and 3.

Due to the changes in the manuscript, I think that the discussion about federal and municipal actions, at least in the context of Brazil and Brazilian networks analysed in the manuscript was diminished in the discussion section of the revised version but it is still in the abstract and also in the introduction. In my opinion, in the discussion it is important to emphasize how the results reveal what is affirmed in the abstract below – which is not so direct - or reduce the emphasis in the last two sentences of the abstract.

Authors: "The municipal and state initiatives are here abstracted as nodes' failures since there is no well defined country-level organization, and the federal actions are the targeted attacks. The results reveal that individual municipalities' initiatives do not cause a high impact on mobility restraint since they tend to be disconnected from the country's global interventions. Oppositely, the coordinated isolation of specific cities is crucial to detach entire network areas and thus prevent spreading. Besides, the targeted attacks pose better results than the reactive strategy."

For instance, to introduce in the abstract the idea that the association is a reasonable ssumption"

Reviewer: "Assuming the municipal ….the targeted attacks, the results reveal that individual ….."

Additional comments

I think that the manuscript may be published in PeerJ after clarifying some points in the letter for reviewers since there are many changes in the revised manuscript that in some features lead to, at least, modified emphasis of the manuscript, in my point of view. Please see the comments introduced in the above sections: basic reporting, experimental deseign and validity of the findings.

Reviewer 2 ·

Basic reporting

The authors have addressed all the points raised. This manuscript can be published in its revised state

Experimental design

The authors have addressed all the points raised. This manuscript can be published in its revised state

Validity of the findings

The authors have addressed all the points raised. This manuscript can be published in its revised state

---

## Round 0.3 · accepted · Accept

Thank you for making the last corrections to the text